# Transfer Learning with Informative Priors: Simple Baselines Better than Previously Reported

**Ethan Harvey**[1*]          *ethan.harvey@tufts.edu*
**Mikhail Petrov**[2*]        *mikhail.petrov@tufts.edu*
**Michael C. Hughes**[1]      *michael.hughes@tufts.edu*
[1]*Department of Computer Science, Tufts University*
[2]*Department of Mechanical Engineering, Tufts University*

**Reviewed on OpenReview:** *https://openreview.net/forum?id=BbvSU02jLg*

## Abstract

We pursue transfer learning to improve classifier accuracy on a target task with few labeled examples available for training. Recent work suggests that using a source task to learn a prior distribution over neural net weights, not just an initialization, can boost target task performance. In this study, we carefully compare transfer learning with and without source task informed priors across 5 datasets. We find that standard transfer learning informed by an initialization only performs far better than reported in previous comparisons. The relative gains of methods using informative priors over standard transfer learning vary in magnitude across datasets. For the scenario of 5-300 examples per class, we find negative or negligible gains on 2 datasets, modest gains (between 1.5-3 points of accuracy) on 2 other datasets, and substantial gains (>8 points) on one dataset. Among methods using informative priors, we find that an isotropic covariance appears competitive with learned low-rank covariance matrix while being substantially simpler to understand and tune. Further analysis suggests that the mechanistic justification for informed priors – hypothesized improved alignment between train and test loss landscapes – is not consistently supported due to high variability in empirical landscapes. We release code[1] to allow independent reproduction of all experiments.

## 1 Introduction

Dataset size is often the limiting factor defining the ceiling of performance in a deep learning application. Without a large dataset of labeled examples available for training, deep learning models face challenges like overfitting and poor generalization (Brigato & Iocchi, 2021). One motivating application with such challenges is medical image classification, especially when labels correspond to diagnoses or annotations obtained by manual review of expert clinicians. In such scenarios, obtaining labels can be prohibitively expensive, and thus often only a small labeled dataset is available (say a few dozen or few hundred examples). Finding other potential sources of information to guide learning can be critical to boost performance.

The field of *transfer learning* offers a promising remedy (Zhuang et al., 2020). The idea is that in addition to a small labeled set for a target task of interest, the user has access to a much larger labeled dataset from a "source" task, such as the well-known ImageNet dataset (Deng et al., 2009). Alternatively, users may have access to a pre-trained model snapshot or distribution derived from such a source. In modern deep learning, the typical off-the-shelf way to benefit from the source task unfolds in two steps (Murphy, 2022, Sec. 19.2). First, obtain a point estimate of neural network weights via "pre-training" on source data. Next, use that point estimate as an initialization for supervised training of the same architecture on the target task, known as "fine-tuning". Throughout, we refer to this two-step process as *standard transfer learning.*

Methods that can deliver further accuracy gains beyond standard transfer learning are naturally of interest to practitioners. Recent results from a new Bayesian transfer learning approach by Shwartz-Ziv et al. (2022) are

---

[*]Equal contribution
[1]Code: https://github.com/tufts-ml/bdl-transfer-learning

particularly exciting. Their approach claims to improve upon standard transfer learning by using the source task to learn a distribution over network weights, not just a point estimate for initialization. This distribution then serves as an *informative prior* for the target task. For the target task of classification on CIFAR-10 data given a training set of just 1000 samples, Shwartz-Ziv et al. (2022) report that their maximum a-posteriori (MAP) objective yields an error rate of 27% compared to the 36% possible with standard transfer learning, a gain in accuracy of 9 percentage points. Similar large gains from MAP with informative priors are reported on other datasets when the target training set size $n$ remains rather small ($n \leq 1000$). Beyond MAP point estimation, they report even further gains from Bayesian estimation of the posterior distribution over weights on the target task via stochastic gradient Hamiltonian Monte Carlo (SGHMC) (Chen et al., 2014).

Excited by the reported gains in the simpler MAP setting, our team undertook an extensive investigation into MAP transfer learning with informative priors. We originally intended to quantify how well the gains that Shwartz-Ziv et al. report on natural image datasets transferred to medical imaging scenarios. However, to our surprise our experiments suggest that **standard transfer learning performs better than previously reported** by Shwartz-Ziv et al. (2022). In systematic experiments that vary the size of the target dataset across four natural image datasets and one medical image dataset from dermatology, we find the relative gains of MAP with informative priors over standard transfer learning vary in magnitude across datasets.

Overall, this paper contributes three findings to the study of transfer learning for point estimated weights:

- First, standard transfer learning without any informative prior performs better than reported in Shwartz-Ziv et al. (2022). In the specific setting of CIFAR-10 with 1000 labeled examples for training and a ResNet-50 architecture (He et al., 2016), we find this method hits test set accuracy between 78-82% across multiple replicate samples of the desired training set size, far better than the 64% accuracy (36% error rate) previously reported. This finding is further supported by third-party experiments in Kaplun et al. (2023), who like us reach accuracy near 80% with a similar architecture in the same setting. Large gains over previous reports are also seen at the $n = 100$ setting (see Fig. 1).

- Second, we find that the relative gains of methods using informative priors over standard transfer learning vary in magnitude across datasets. For the scenario of 5-300 examples per class, we find negative or negligible gains on 2 datasets, modest gains (between 1.5-3 points of accuracy) on 2 other datasets, and substantial gains (>8 points) on one dataset. Among methods using informative priors, we find that Chelba & Acero (2006)'s isotropic covariance appears competitive with Shwartz-Ziv et al. (2022)'s low-rank covariance matrix while being substantially simpler to understand and tune.

- Finally, we empirically assess a specific mechanistic hypothesis from Shwartz-Ziv et al. (2022), who suggest informative priors allow improved "alignment" between the train and test loss landscape of the target fine-tuning task. For an idealized illustration, see their Figure 1. We find that on real datasets, informative priors do not often align train and test losses well; instead our Fig. 2 shows there is considerable variability in the alignment especially across samples of a small labeled set from the target task.

To take steps to make our present analysis reproducible (including all dataset splitting and hyperparameter tuning procedures), we release a codebase[1] that can re-run all experiments and reproduce all figures here. In the spirit of open science, we welcome any feedback or questions from users.

Our inability to reproduce previously reported results is part of a larger trend across machine learning research (Pineau et al., 2021). We hope our work here helps the community move toward improved best practices. The first best practice we advocate is sharing scripts that enable all experiments, especially those related to hyperparameter tuning. The second best practice is allowing all baselines at minimum the same hyperparameter tuning as a presented method in every experimental setting.

## 2 Background

**Bayesian transfer learning.**  Recent work by Shwartz-Ziv et al. (2022) proposed Bayesian transfer learning, where a re-scaled posterior from the source task is used as the prior for the target task. This work is motivated

by sequential Bayesian updating, where some source data $\mathcal{D}_S$ is acquired, a posterior $p(w|\mathcal{D}_S)$ over weights $w$ is formed, and this posterior is used as an *informed prior* for a target dataset $\mathcal{D}_T$ (Murphy, 2012)

$$p(w|\mathcal{D}_S) \propto p(\mathcal{D}_S|w)p(w) \tag{1}$$
$$p(w|\mathcal{D}_T, \mathcal{D}_S) \propto p(\mathcal{D}_T|w)p(w|\mathcal{D}_S).$$

In sequential Bayesian updating, analysts often assume all data is independent and identically distributed, at least when conditioned on parameters $w$. However, in transfer learning the target data comes from a somewhat different distribution than the source data, so beliefs about which weights are a-priori plausible may need adjustment. To rectify this problem, Shwartz-Ziv et al. (2022) introduce a scaling factor $\lambda \geq 1$ controlling influence of the source-informed prior over weights $w$. Concretely, $\lambda$ rescales the covariance matrix of the posterior $p(w|\mathcal{D}_S)$ (which they assume is well-approximated by a Gaussian) to form the source-informed prior. When $\lambda = 1$, the model directly uses the source posterior as the source-informed prior. As $\lambda$ grows larger, the source-informed prior has inflated covariance. In the limit as $\lambda \to \infty$ the influence of the source data disappears (the prior approaches a uniform distribution) and the only influence of the source task on the target task would come from initialization, thus recovering standard transfer learning.

## 2.1 Common framework for MAP transfer learning.

**Probabilistic model for target task.** Following Shwartz-Ziv et al. (2022), we form a model that performs a desired target task of image classification with $C$ possible classes. The model has two parts: a backbone and a classification head. Let $f$ denote the backbone neural network with weights $w \in \mathbb{R}^d$. For modern computer vision, $d$ might be in the millions or even billions. Given an image $x_i$ of prespecified size, $f$ produces a hidden representation vector $f_w(x_i) \in \mathbb{R}^H$. We assume the first entry of this vector is always 1 to allow learning intercepts. We define the classifier head as parameterized by weights $V \in \mathbb{R}^{C \times H}$ that can map the representation vector to logits over $C$ classes. Then, given a target training set of size $n$, containing pairs $\{x_i, y_i\}_{i=1}^n$ of images $x_i$ and categorical labels $y_i \in \{1, 2, \ldots C\}$, we have the following **template** for a probabilistic model:

$$p(w) = \mathcal{N}(w \mid \mathbf{m}, \lambda\mathbf{S}) \qquad \text{source-informed prior on backbone} \tag{2}$$
$$p(V) = \mathcal{N}(\text{vec}(V) \mid 0, \tau I), \qquad \text{prior on clf. head} \tag{3}$$
$$p(y_{1:n}|x_{1:n}, w, V) = \prod_{i=1}^n \text{Cat}\big(y_i|\text{softmax}(Vf_w(x_i))\big) \qquad \text{likelihood} \tag{4}$$

Here, $\tau > 0$ and $\lambda > 0$ are key hyperparameters that require tuning, while the choice of $\mathbf{m}, \mathbf{S}$ is method specific; these values may be simple (e.g. an all-zero mean) or informed by a source-task.

**Target task MAP estimation.** We can fit the above model to the target dataset via a MAP point estimation strategy, finding values of weights $w, V$ that minimize the objective

$$L(w, V) := -\frac{1}{n}\Big[\sum_{i=1}^n \log p(y_i|x_i, w, V) + \log p(w) + \log p(V)\Big] \tag{5}$$

Given a concrete target dataset, we pursue stochastic first-order gradient descent to obtain estimates of $w, V$. In transfer learning, it is common to initialize backbone weights $w$ at a known good value from the source task, denoted $\mu$. Classifier head weights $V$ can be initialized randomly.

As surveyed below, several approaches to transfer learning in the literature share both the common model template defined above and the pursuit of MAP estimation. They vary only in the setting of the backbone prior's mean vector $\mathbf{m}$ and covariance matrix $\mathbf{S}$. Tab. 1 summarizes the specific possible methods.

## 2.2 Concrete methods for MAP transfer learning.

Below we cover a spectrum of possible choices for the prior over backbone weights, all of which can be integrated into the common MAP transfer learning framework above. They differ in whether and how the prior is informed by a source task, especially the source-informed point estimate $\mu$ of the backbone weights.

**Standard non-informed isotropic prior.** The *standard* transfer learning approach most common in the literature is to set $\mathbf{m}$ to the all zero vector and set $\mathbf{S}$ to the identity matrix. We refer to this as *StdPrior* throughout (read as "standard prior"). Here, the backbone prior is not informed by the source task; the only source-informed knowledge comes from initializing MAP estimation at source-informed weights $\mu$.

Table 1: Possible methods for point estimation of neural network weights for a target task. All can be seen as MAP estimation for model defined by a common likelihood and a method-specific prior. The MAP objective is then optimized by first-order stochastic gradient descent. Our work focuses specifically on methods informed by a source task (ImageNet classification), using values of weight vector $\mu$ and low-rank (LR) covariance matrix $\Sigma$ *learned* from this pre-training phase. We include columns for related works and mark the methods they investigate.

| Method | also known as | Prior | Init. | Shwartz-Ziv et al. (2022) | Špendl & Pirc (2023) | ours |
|---|---|---|---|---|---|---|
| StdPrior fromScratch | SGD Non-Learned Prior in Shwartz-Ziv et al. | $\mathcal{N}(0, \lambda I)$ | random | ✓ | ✓ | ✗ |
| StdPrior fromImgNet | SGD Transfer Init in Shwartz-Ziv et al. | $\mathcal{N}(0, \lambda I)$ | $\mu$ | ✓ | ✗ | ✓ |
| LearnedPriorIso fromImgNet | MAP adaptation in Chelba & Acero | $\mathcal{N}(\mu, \lambda I)$ | $\mu$ | ✗ | ✗ | ✓ |
| LearnedPriorLR fromImgNet | SGD Learned Prior in Shwartz-Ziv et al. | $\mathcal{N}(\mu, \lambda \Sigma)$ | $\mu$ | ✓ | ✓ | ✓ |

**Learned prior with isotropic covariance.** A natural way to let the source task inform the backbone prior is to set the mean such that $\mathbf{m} = \mu$. The prior covariance $\mathbf{S}$ is left as the identity matrix. This corresponds to an older method from Chelba & Acero (2006), known as *MAP adaptation*, which was recommended as the "standard baseline for solving transfer learning tasks and benchmarking new algorithms" in Xuhong et al. (2018). We refer to this as *LearnedPriorIso* throughout.

**Learned prior with low-rank (LR) covariance.** Shwartz-Ziv et al. (2022) propose informing both the prior's mean and covariance matrix from the source task. They use Stochastic Weight Averaging-Gaussian (SWAG) (Maddox et al., 2019) to approximate the posterior distribution $p(w|\mathcal{D}_S)$ of the source data $\mathcal{D}_S$ with a Gaussian distribution as $\mathcal{N}(\mu, \Sigma)$ where $\mu$ is the learned mean and $\Sigma = \frac{1}{2}(\Sigma_{\text{diag}} + \Sigma_{\text{LR}})$ is a representation of a covariance matrix with both diagonal and *low-rank* components. For a $d$-parameter neural net with typical $d$ in millions or billions, a standard $d \times d$ covariance matrix is unaffordable to store. Thus, we select a concrete desired rank $k$ (where $k \ll d$), and form the LR covariance as $\Sigma_{\text{LR}} = \frac{1}{k-1}QQ^T$, where $Q \in \mathbb{R}^{d \times k}$ is a learned parameter from the source task (the diagonal component $\Sigma_{\text{diag}}$ is also learned). We refer to this prior as *LearnedPriorLR* throughout, and use $k = 5$ throughout as recommended by Shwartz-Ziv et al..

### 2.3 Previous replication efforts with informative priors.

Work on informative priors by Shwartz-Ziv et al. (2022) has generated considerable follow-up attention, including a previous replication study with different goals than ours. Špendl & Pirc (2023) sought to compare "from scratch" training on the target task alone (without even a source-informed initialization) to the proposed informed prior MAP procedure. Špendl & Pirc (2023) used Shwartz-Ziv et al. (2022)'s released code with a fixed initial learning rate and weight decay on just one dataset (Oxford Flowers-102 dataset by Nilsback & Zisserman (2008)). They report that transfer learning with informed priors did outperform a simple from-scratch baseline that did not do *any* transfer from source to target (see Tab. 1 for a comparison of which methods all papers investigate). In contrast, we run experiments designed to compare standard initialization-only transfer learning to informed prior transfer learning. Our experiments also perform separate hyperparameter tuning for all methods at all dataset sizes.

## 3 Experimental Procedures

We seek to fairly compare three distinct methods for point estimating optimal neural network weights $w, V$ for a target classification task. All three make use of pre-training on a source task, some via initialization only and some via an additional learned prior, as described in Sec. 2.2 and summarized in Table 1. We refer to these methods in all result tables and figures as *StdPrior* for standard transfer learning with an uninformative standard Gaussian prior with zero mean and isotropic covariance $\mathbf{S} = \lambda I$, *LearnedPriorIso* for a source-informed prior with learned mean $\mu$ but isotropic covariance $\mathbf{S} = \lambda I$, and *LearnedPriorLR* for the source-informed prior with mean $\mu$ and learned low-rank covariance multiplied by scalar $\lambda$.

Below we provide essential details of our experimental procedure. We intend that all experiments are fully reproducible by others. See Tab. B.1 in App. B for a detailed comparison of Shwartz-Ziv et al. (2022)'s and our experimental procedures. For further details, see App. A as well as our released codebase.

**Datasets and classification tasks.** We study targeted classification tasks from five distinct open-access datasets. For all tasks, we study how well variants of MAP estimation can *transfer* knowledge from a source task on ImageNet.

The first three datasets were selected as common benchmarks that enable direct comparisons to previous experiments by Shwartz-Ziv et al. (2022): the 10-way classification task of CIFAR-10 (Krizhevsky et al., 2010), the 102-way classification task of Oxford Flowers (Nilsback & Zisserman, 2008), and the 37-way classification task of Oxford-IIIT Pets (Parkhi et al., 2012). Each of these contains natural images (RGB, resized to 224x224 pixels) that typically depict one object of the labeled class.

For the last two datasets, we select the 100-way classification task of Fine-Grained Visual Classification of Aircraft (FGVC-Aircraft) (Maji et al., 2013) and HAM10000 (Tschandl et al., 2018; Codella et al., 2019), an open-source deidentified dataset, informally known as "Skin Cancer MNIST". HAM10000 was designed to support assessment of automatic diagnosis of several possible skin lesion types from RGB images resized to 224x224 pixels of skin surface microscopy (dermatoscopy). To avoid extreme imbalance, we form a 4-class classification task between 3 original classes with sufficient data (at least 10% prevalence) and an additional "other" class that combines all other original classes.

**Common architecture and optimization.** Across all experiments, we always set the backbone architecture of network $f$ to a ResNet-50 (He et al., 2016). This directly matches the experiments in Shwartz-Ziv et al. (2022). Our PyTorch-based codebase uses a common optimization procedure for training: SGD with batch size of 128 and Nesterov momentum of 0.9 with cosine-annealing schedule for learning rate (Loshchilov & Hutter, 2016). Initial learning rate is a tuned hyperparameter (see below).

**Common source-task knowledge.** For all methods, we take the learned weights $\mu$ that form the initialization (and if needed, the prior mean), from Shwartz-Ziv et al. (2022)'s released snapshots[2]. For *LearnedPriorLR*, we also obtain the diagonal and low-rank ingredients for the covariance matrix directly from Shwartz-Ziv et al. (2022)'s released snapshots. We specifically use Shwartz-Ziv et al.'s snapshots pre-trained on ImageNet with SimCLR (Chen et al., 2020), following their recommendation that pre-training with self-supervised learning (as in SimCLR) leads to improved transfer learning compared with fully-supervised learning (e.g. using cross-entropy loss with ImageNet class labels).

**Fixes to Shwartz-Ziv et al. (2022)'s code.** Shwartz-Ziv et al. (2022)'s released code performs inconsistent scaling by $\lambda$ of the low-rank and diagonal components of the covariance matrix of their informed prior. Looking at their code (see line 40 of `utils.py`), they obtain $\Sigma_{\text{LR}}$ by rescaling the low-rank matrix $Q$ by $\lambda$ before computing $QQ^T$, which yields an overall covariance matrix $\frac{\lambda}{2}\Sigma_{\text{diag}} + \frac{\lambda^2}{2}\Sigma_{\text{LR}}$. To remedy this, we rescale the low-rank matrix $Q$ by $\sqrt{\lambda}$ instead of $\lambda$, obtaining the intended $\frac{\lambda}{2}\Sigma_{\text{diag}} + \frac{\lambda}{2}\Sigma_{\text{LR}}$. We also remove the redundant "double prior" due to their implementation's use of weight decay on weights $w$ in addition to the informative prior. Details for both of these changes can be found in Tab. B.1 in App. B.

**Varying train set size $n$.** For each dataset, we setup a range of transfer learning tasks by artificially varying $n$, the number of available samples for training. In all cases, we ensure *balanced class frequencies*, ensuring the same number of images per class. For CIFAR-10, by selecting $n \in \{10, 100, 1000, 10000, 50000\}$, we cover exactly 1, 10, 100, 1000, and all available examples per class. For Oxford Flowers, we select $n \in \{102, 510, 1020\}$ to cover 1, 5, and 10 examples per class. For Oxford-IIIT Pets, we select $n \in \{37, 370, 3441\}$ to cover 1, 10, and 93 examples per class (93 is the maximum possible value that keeps exact balance). For FGCV-Aircraft, we select $n \in \{100, 500, 1000, 5000\}$ to cover 1, 5, 10, and 50 examples per class. For HAM10000, we select $n \in \{100, 1000\}$. We emphasize that the 10-1000 examples per class regime is likely of most practical interest for many applications.

**Repeated trials.** At each chosen value of $n$, we build 3 separate "replicate" training sets by randomly sampling $n$ image-label pairs (without replacement) from the predefined complete training set. We then separately perform all training (including hyperparameter tuning) on each replicate. We emphasize we are careful to ensure similar class-composition for all replicates.

---

[2]Shwartz-Ziv et al. (2022)'s SimCLR snapshots can be found at `https://github.com/hsouri/BayesianTransferLearning`

**Hyperparameter tuning.** All methods need to tune three key hyperparameters: the covariance scaling factor $\lambda$, the classification head weight decay $\tau$, and the initial learning rate. To give informative prior approaches the best possible settings, for *LearnedPriorLR* we tune all three together. For simplicity, for both *StdPrior* and *LearnedPriorIso* we fix $\lambda = \tau$, tuning only $\tau$ and learning rate. In some ways, this setup puts standard transfer learning at a disadvantage, as *LearnedPriorLR* is allowed to search a larger hyperparameter space.

Model fitting (including tuning) has two stages. In the first stage, we first randomly divide available data of size $n$ into a train set and validation set with 4:1 ratio. We then tune all hyperparameters via grid search (ranges in App. A.2), selecting the best configuration based on the log likelihood of the labels of the validation set. After selecting an optimal hyperparameter configuration in stage one, we refit to the entire dataset of $n$ examples in stage two. In all stages, the optimizer runs until reaching a specified maximum number of iterations.

**Performance metrics on test set.** All datasets provide a predefined train/test split which our experiments respect. We report heldout performance metrics on the complete available test set for each task. For CIFAR-10, Oxford Flowers, Oxford-IIIT Pets, and FGVC-Aircraft, we report *classifier accuracy* (higher is better) following Shwartz-Ziv et al. (2022). For HAM10000, we report *average AUROC* (higher is better), macro-averaged over all classes. Ultimately, we report the average performance metric across all 3 replicate trials, as well as the min-max range. For all tasks, we also assess the *negative log likelihood* (NLL) of the test set.

## 4 Results

Examining the results of all experiments, we come to three key findings. These are detailed in the subsections below.

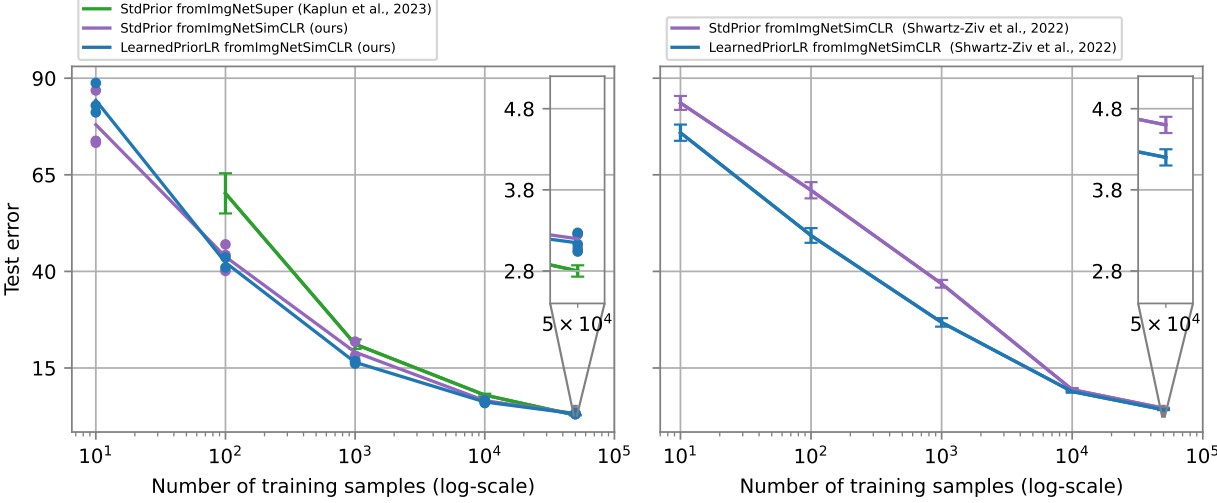

Figure 1: Error rate (lower is better) vs. target train set size on CIFAR-10, for various MAP estimation methods for transfer learning from ImageNet. *Left:* Our results. *Right:* Results copied from Shwartz-Ziv et al. (2022) (their Tab. 10). **Takeaway: In our experiments, standard transfer learning (*StdPrior*) does better than previously reported.** *Setting details:* The blue and purple lines across both panels come from comparable settings: a common ResNet-50 architecture and common learned values for mean and low-rank (LR) covariance taken directly from the SimCLR pre-trained snapshots in Shwartz-Ziv et al. (2022)'s repository. *Green line:* The left panel's green line is a third-party experiment copied from Kaplun et al. (2023), suggesting others can achieve similar performance as we do for standard transfer learning with ResNet-50. They use fully-supervised pre-training not self-supervised SimCLR. Plotted mean and standard deviations confirmed via direct correspondence with Kaplun et al..

Table 2: CIFAR-10 heldout accuracy (higher is better) as target train set size $n$ increases. For each method and train set size $n$, we ran our two-phase training pipeline (including hyperparameter tuning) on 3 distinct training sets (independent samples of $n$ examples from the provided full training set). We report mean accuracy on test set across these 3 trials, along with the min-max range. 10 classes, balanced. See App. A for details.

| Method | $n = 10$ (1/cl.) | 100 (10/cl.) | 1000 (100/cl.) | 10000 (1k/cl.) | 50000 (5k/cl.) |
|---|---|---|---|---|---|
| StdPrior fromImgNet | 22.0 (13.2-26.7) | 56.2 (53.0-59.9) | 80.9 (78.2-82.8) | 93.4 (93.3-93.7) | 96.8 (96.7-96.9) |
| LearnedPriorIso fromImgNet | 20.7 (17.7-23.4) | 56.7 (54.8-59.8) | 81.9 (81.6-82.1) | 94.3 (94.1-94.5) | 97.3 (97.2-97.4) |
| LearnedPriorLR fromImgNet | 15.7 (11.3-18.8) | 58.7 (56.4-60.6) | 83.5 (83.3-83.9) | 93.8 (93.4-94.1) | 96.9 (96.7-97.0) |

Table 3: Oxford Flowers heldout accuracy (higher is better) as target train set size $n$ increases. For each method and train set size $n$, we ran our two-phase training pipeline (including hyperparameter tuning) on 3 distinct training sets (independent samples of $n$ examples from the provided full training set). We report mean accuracy on test set across these 3 trials, along with the min-max range. 102 classes, balanced. See App. A for details.

| Method | $n = 102$ (1/cl.) | 510 (5/cl.) | 1020 (10/cl.) |
|---|---|---|---|
| StdPrior fromImgNet | 31.1 (12.6-40.6) | 78.9 (78.4-79.3) | 87.9 (87.4-88.1) |
| LearnedPriorIso fromImgNet | 19.2 ( 9.0-34.8) | 79.1 (78.5-79.4) | 88.6 (88.2-89.1) |
| LearnedPriorLR fromImgNet | 28.8 (10.6-39.2) | 77.9 (74.8-79.4) | 88.4 (88.2-88.7) |

Table 4: Oxford-IIIT Pets heldout accuracy (higher is better) as target train set size $n$ increases. For each method and train set size $n$, we ran our two-phase training pipeline (including hyperparameter tuning) on 3 distinct training sets (independent samples of $n$ examples from the provided full training set). We report mean accuracy on test set across these 3 trials, along with the min-max range. 37 classes, balanced. See App. A for details.

| Method | $n = 37$ (1/cl.) | 370 (10/cl.) | 3441(93/cl.) |
|---|---|---|---|
| StdPrior fromImgNet | 17.3 (14.5-20.0) | 54.8 (53.2-57.5) | 86.4 (86.0-86.6) |
| LearnedPriorIso fromImgNet | 7.0 ( 5.4- 8.5) | 55.5 (53.3-57.7) | 86.4 (85.4-87.0) |
| LearnedPriorLR fromImgNet | 6.6 ( 6.2- 7.3) | 57.4 (56.2-58.2) | 86.7 (85.0-87.8) |

## 4.1 Finding 1: Standard transfer learning better than reported in Shwartz-Ziv et al. (2022).

Our Fig. 1 provides a side-by-side comparison of results on CIFAR-10 for our experiments (left) compared to those originally reported in Shwartz-Ziv et al. (2022) (right). Here, we report error rate (lower is better) and use a color scheme that matches original plots from Shwartz-Ziv et al.. We see that *StdPrior* (purple line) is consistently 10-15 percentage points better at dataset sizes $n \in \{100, 1000\}$, essentially erasing the gap in performance previously attributed to informative priors.

Our results are further supported by other third-party experiments by Kaplun et al. (2023) that use the same architecture (ResNet-50) and same source task (ImageNet) to classify CIFAR-10. Our results at $n = 1000$ show test error for *StdPrior* in the 17-22% range, matching results from Kaplun et al. (green line in our figure) well. We suggest this lends credibility to our claim that it is possible for others to do far better with standard transfer learning than has been previously reported. We note that Kaplun et al. do pre-training with supervised methods, not the self-supervised SimCLR; see App. D for a comparison with Shwartz-Ziv et al. (2022)'s fully-supervised pre-training results.

## 4.2 Finding 2: Relative gains of informed priors over standard transfer learning vary across datasets.

Ultimate test set performance for all methods across train set sizes $n$ is reported for all 5 datasets in Tables 2, 4, 3, 5, and 6 respectively. Each table entry reports that method's mean test accuracy across the 3 replicate samples of $n$ training points, as well as the min-max range.

We find that the relative gains of methods using informative priors over standard transfer learning vary in magnitude across datasets. For the scenario of 5-300 examples per class, we find negative or negligible gains on Oxford Flowers and HAM10000; modest gains (between 1.5-3 points of accuracy) on CIFAR-10 and Oxford-IIIT Pets; and substantial gains (>8 points) on FGVC-Aircraft.

Among methods using informative priors, we find that Chelba & Acero (2006)'s isotropic covariance appears competitive with Shwartz-Ziv et al. (2022)'s low-rank covariance matrix while being substantially simpler to understand and tune. There are a few cases where Shwartz-Ziv et al. (2022)'s low-rank covariance

Table 5: FGVC-Aircraft heldout accuracy (higher is better) as target train set size $n$ increases. For each method and train set size $n$, we ran our two-phase training pipeline (including hyperparameter tuning) on 3 distinct training sets (independent samples of $n$ examples from the provided full training set). We report mean accuracy on test set across these 3 trials, along with the min-max range. 100 classes, balanced. See App. A for details.

| Method | $n = 100$ (1/cl.) | 500 (5/cl.) | 1000 (10/cl.) | 5000 (50/cl.) |
|---|---|---|---|---|
| StdPrior fromImgNet | 2.6 (1.3-4.6) | 22.5 (21.4-24.4) | 40.6 (39.8-41.8) | 85.7 (85.4-86.0) |
| LearnedPriorIso fromImgNet | 3.5 (2.8-4.5) | 25.9 (24.0-27.3) | 51.3 (50.0-52.6) | 85.9 (85.8-86.0) |
| LearnedPriorLR fromImgNet | 3.8 (3.5-4.2) | 24.5 (23.7-25.3) | 50.9 (50.4-51.9) | 84.2 (83.5-85.3) |

Table 6: HAM10000 AUROC on heldout data (higher is better, macro-averaged across classes) as target train set size $n$ increases. For each method and size $n$, we ran our two-phase training (including hyperparameter tuning) on 3 distinct training sets (independent samples of size $n$ from the provided full training set). We report mean AUROC on test set across 3 trials (min-max range). 4 classes, ranging from 11%-67% prevalance. See App. A for details.

| Method | $n = 100$ | 1000 |
|---|---|---|
| StdPrior fromImgNet | 78.1 (75.0-82.8) | 85.6 (85.0-86.3) |
| LearnedPriorIso fromImgNet | 78.0 (75.0-82.8) | 86.5 (85.5-87.4) |
| LearnedPriorLR fromImgNet | 78.7 (74.9-82.7) | 86.6 (85.0-87.5) |

is clearly better than Chelba & Acero (2006)'s isotropic covariance: CIFAR-10 at $n$=100 has 58.7% for *LearnedPriorLR* vs. 56.7% for *LearnedPriorIso*; CIFAR-10 at $n$=1000 has 83.5% for *LearnedPriorLR* vs. 81.9% for *LearnedPriorIso*; and Oxford-IIIT Pets at $n$=370 has 57.4% accuracy for *LearnedPriorLR* vs. 55.5% accuracy for *LearnedPriorIso*. However, these wins are not consistent across datasets, and when there are substaintal gains from informed priors, Chelba & Acero (2006)'s isotropic covariance is competitive with Shwartz-Ziv et al. (2022)'s low-rank covariance (see FGVC-Aircraft at dataset sizes $n \in \{500, 1000\}$).

Further results in App. C compare methods using NLL as a performance metric. Again, the conclusion is that the relative gains of methods using informative priors over standard transfer learning vary in magnitude across datasets.

### 4.3 Finding 3: Large variability in quality of alignment between training and test loss landscapes.

Shwartz-Ziv et al. (2022) suggest an underlying mechanism for why informed priors improve transfer learning: better alignment between the loss landscapes of the target classification task across the small available training set and the large test set. (Remember, test set performance is assumed a reliable "gold-standard" here due to its size, but such data is not available for training in the proposed transfer learning setting). See an idealized illustration in their Figure 1, where the overall function shape and especially the locations of minima are far more similar across train and test with source-informed priors than standard methods.

To empirically study this proposed phenomenon, we examine how informed priors reshape the loss surface on the CIFAR-10 task when $n = 1000$. It is difficult to visualize the high-dimensional input space (the size of all weights $w$ is over 23 million), so we look at a 1D slice that linearly interpolates between two values: the optima $w^*_{Std}$ found by *StdPrior* MAP estimation or the optima $w^*_{LR}$ found by *LearnedPriorLR* and the optima $w^*$ found at the largest dataset size. (We interpolate between classifier heads $V$ as well, but use just $w$ symbols for simplicity). Each panel of Fig. 2 visualizes over this 1D interpolation how two functions behave: the train set loss of each MAP objective (red line, vary by columns as *StdPrior* or *LearnedPriorLR*), and the test set NLL (blue, vary by columns as different 1D slices). Each row comes from a different replicate sample of $n = 1000$ image-label pairs for training.

If informed priors lead to better alignment, we contend that this should be visible in these plots: the location of the minima for the blue test curve should be closer to the source-informed optima $w^*_{LR}$ than the standard method optima $w^*_{Std}$. We label the Euclidean distance in each plot to help readers focus on which method reduces the gap between trained and ideal minimum. However, across 3 replicate train sets (rows) we find there is high variability in how methods rank in alignment. Sometimes, $w^*_{LR}$ does have a smaller gap, indicating better alignment (bottom row). However, we also find the gap can be similar (middle row) or even prefer the standard method (top row). We thus suggest that there is considerable variability in alignment quality, making the empirical evidence for the loss-alignment hypothesis of Shwartz-Ziv et al. (2022) less decisive than their idealized illustration would suggest.

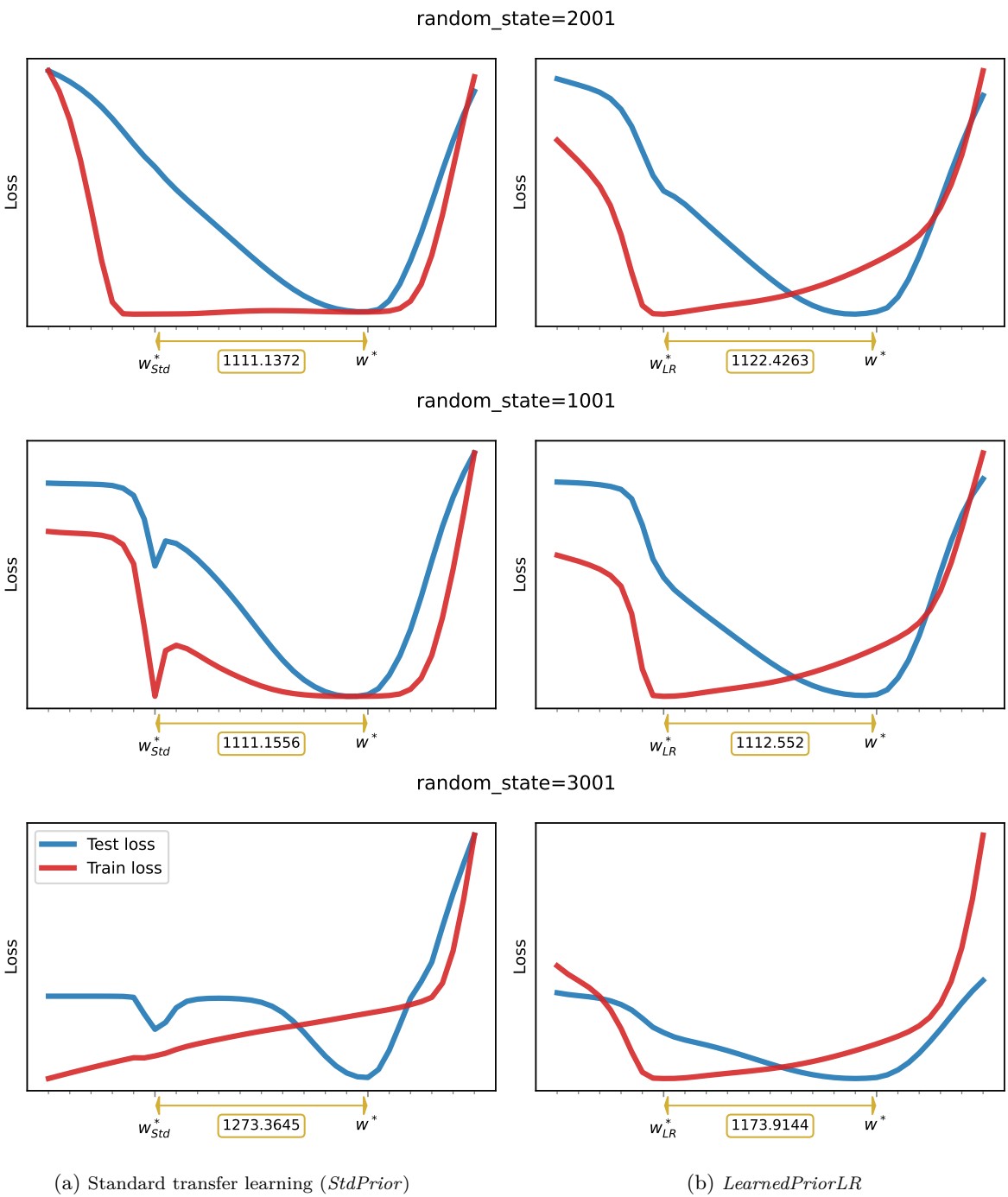

(a) Standard transfer learning (*StdPrior*)         (b) *LearnedPriorLR*

Figure 2: Empirical alignment of loss landscapes for target task across train and test for CIFAR-10 $n = 1000$. Compare to Shwartz-Ziv et al.'s Figure 1, which is an idealized illustration not an empirical result. *Each panel:* We assess a 1D slice of the high-dimensional landscape by linearly interpolating parameters $w$ between the optima $w^*_{Std}$ found via minimizing the standard MAP objective (left) or the optima $w^*_{LR}$ found via minimizing the *LearnedPriorLR* MAP objective (right) and the optima $w^*$ found at the largest dataset size. Red curve shows the indicated training loss (varies by column) on CIFAR-10 with $n = 1000$ samples; blue curve (varies by column) shows CIFAR-10 test set NLL. Each row shows results from a different train set sample of size $n = 1000$. The gap between the optima found via training and the test set's ideal minimum is shown as a double-sided gold arrow. **Takeaway: Shwartz-Ziv et al. (2022)'s learned prior approach does not always reduce the gap between trained and ideal minimum.**

## 5    Discussion and Conclusion

Identifying the reason our current study leads to different conclusions than prior work is complex. The code released by the original study authors was incredibly useful to us. We are very thankful to the authors for making it available, it was more usable than other code repositories we have seen. However, we found their released code does not allow exact replication of key experimental steps from Shwartz-Ziv et al. (2022), such as replicable sampling of the same $n$ image-label pairs they used for training in their experiments, or how exactly to perform hyperparameter tuning (how to carve out a validation set from available data, etc.). It is thus difficult to isolate what setting in our present study lead to the different results seen here, though we attempt a careful side-by-side comparison in App. B. Our current best guess is that our substantial effort into fair hyperparameter tuning of all methods led to improved numbers for standard transfer learning.

**Is standard transfer learning better than full Bayesian inference?**    We stress that all findings here are confined to the simpler MAP point estimation setting; we do not study the full Bayesian posterior estimation methods of Shwartz-Ziv et al. (2022), which require notable additional computational complexity across fine-tuning and testing phases of the model development life cycle. For example, fine-tuning is usually at least 5-10x as time-consuming. Full Bayesian approaches also require substantive expertise from downstream practitioners when debugging is needed.

That said, our standard transfer learning results on CIFAR-10 are as good as or better than the full Bayesian inference results reported in Shwartz-Ziv et al. (2022) (for a direct comparison, see Fig. F.1 in App. F). We expect deep ensembles to further improve on our results (Lakshminarayanan et al., 2017). We leave it to future work to investigate the relative performance gains from full Bayesian inference using source-informed priors compared to properly tuned deep ensemble baselines.

**What transfer learning method is recommended?**    Based off our experiments, when point estimating neural network weights we recommend both *StdPrior* and *LearnedPriorIso* for their simplicity and performance. We do not see enough of a large consistent boost in discriminative performance from Shwartz-Ziv et al. (2022)'s low-rank covariance to clearly justify its downstream use as a "default" approach. However, there are a few cases at small dataset sizes where Shwartz-Ziv et al. (2022)'s low-rank covariance is clearly better than Chelba & Acero (2006)'s isotropic covariance and may be a method to reach for to boost target task performance by a couple percentage points.

**Limitations.**    We only consider transfer learning from one fixed source task, ImageNet. Further work could investigate if informative priors may be more useful if the source task is more directly related to a target task of interest. To define the priors, we also only consider one fixed mean vector and learned low-rank (with rank $k = 5$) covariance matrix taken directly from released snapshots by others. Others could find different behavior with other pre-training procedures, especially with more tuning of the rank on the target task.

**Outlook.**    We stress that our ultimate goal is to improve the community's shared understanding about how to best achieve the goals of transfer learning, not to cast doubt on a specific past study. We hope that moving forward, two best practices will become more widespread. First, sharing code that allows reproducing all experimental results (including dataset preparation and hyperparameter tuning). Second, allowing all methods (especially baselines) the same time and compute resources for hyperparameter tuning.

### Acknowledgments

Effort by authors EH and MCH was supported in part by a grant from the Alzheimer's Drug Discovery Foundation. Computing infrastructure support provided in part by the U.S. National Science Foundation under award OAC CC* 2018149.

We thank Ravid Shwartz-Ziv, Micah Goldblum, and Hossein Souri for answers to our questions, as well as making their codebase available. We also thank anonymous reviewers for suggestions that substantially improved the content and presentation of our paper.

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

# A  Classification

## A.1  Dataset Details

Our experiments include a replication of CIFAR-10 (Krizhevsky et al., 2010) experiments from Shwartz-Ziv et al. (2022), similar Oxford Flowers (Nilsback & Zisserman, 2008) experiments to Shwartz-Ziv et al. (2022), similar Oxford-IIIT Pets (Parkhi et al., 2012) experiments to Shwartz-Ziv et al. (2022), additional FGVC-Aircraft (Maji et al., 2013) experiments, and additional HAM10000 (Tschandl et al., 2018; Codella et al., 2019) experiments.

For Oxford Flowers experiments, we modify prior experiments to enforce classes are uniformly distributed in the training data, that at least one sample from each class is included in all tested values of $n$, and that the validation set is proportional with the training set. Shwartz-Ziv et al. (2022) validate on the predefined validation set specified by the dataset creators. In our minds, it is not realistic to train on a range of dataset sizes of 102 or 510 images but validate on 1020 images. Practitioners looking for guidance in how to use transfer learning on their dataset will be unlikely to put more images in validation than training. Instead, we ignore the predefined validation set and use 1/5 of the training set for our two-phase training pipeline.

For Oxford-IIIT Pets experiments, we modify prior experiments to enforce classes are uniformly distributed in the training data, and that at least one sample from each class is included in all tested values of $n$.

For FGVC-Aircraft experiments, we combine the training and validation sets specified by the creators to enforce that the validation set is proportional to the training set.

For HAM10000 experiments, we form a classification task using classes with sufficient data (at least 10% prevalence). One additional "other" class was formed from all other classes that did not have sufficient

Table A.1: Distribution of labels in HAM10000 training set.

| Label | Percent of training set |
|---|---|
| Melanocytic | 66.95% |
| Melanoma | 11.11% |
| Benign keratosis | 10.97% |
| Other | 10.97% |

prevalance. For the validation set, we ensure each patient's data belongs to exactly one split and stratify by class to ensure comparable class frequencies.

We use the same preprocessing steps for all five datasets. For each distinct training set size $n$, we compute the mean and standard deviation of each channel to normalize images. During fine-tuning we resize the images to 256x256 pixels, perform random cropping to 224x224, and perform horizontal flips. At test time, we resize the images to 256x256 pixels and center crop to 224x224.

## A.2 Classifier Details

We use SGD with a Nesterov momentum parameter of 0.9 for optimization. We train for 6,000 steps using a cosine annealing learning rate (Loshchilov & Hutter, 2016). We select the initial learning rate from 4 logarithmically spaced values between 0.1 to 0.0001 and the weight decay from 5 logarithmically spaced values between $1e^{-2}$ to $1e^{-6}$, as well as without weight decay. For *LearnedPriorLR*, we select the covariance scaling factor from 10 logarithmically spaced values between $1e^0$ to $1e^9$.

While tuning hyperparameters, we hold out 1/5 of the training set for validation, ensuring balanced class frequencies between sets. At the smallest tested values of $n$ suggested by Shwartz-Ziv et al. (2022), the number of training samples is equal to the number of classes, and thus any images assigned to the validation set will have their corresponding class not represented in the training data.

After selecting the optimal hyperparameters from the validation set NLL, we retrain the model using the selected hyperparameters on the combined set of all $n$ images (merging training and validation). All results report performance on the task in question's predefined test set.

***StdPrior.*** Stocastic gradient descent (SGD) in PyTorch (Paszke et al., 2019) implements weight decay via an efficient direct modification of the gradient vector. Given decay parameter $\alpha > 0$, at iteration $t$ the optimizer internally performs an additive incremental update of the gradient with respect to backbone weights: $\nabla_w \ell(w_{t-1}) = \nabla_w \ell(w_{t-1}) + \alpha w_{t-1}$ where $\ell(w)$ is the (unpenalized) loss function (e.g. cross entropy). The gradient with respect to classifier parameters $V$ is updated similarly. This direct modification is equivalent to, but more efficient than, automatic differentiation of a penalized loss that includes an L2 penalty term for each of $w$ and $V$:

$$L(w,V) := \underbrace{-\frac{1}{n}\sum_{i=1}^{n}\log p(y_i|x_i,w,V)}_{\ell(w,V)} + \frac{\alpha}{2}||w||^2 + \frac{\alpha}{2}||\text{vec}(V)||^2. \tag{6}$$

Weight decay in PyTorch has a Bayesian interpretation as maximum a-posteriori (MAP) estimation of parameters $w, V$, under the assumption that the weights $w, V$ each follow a Gaussian prior $\mathcal{N}(0, \tau I)$ with scalar precision parameter $\tau = \frac{1}{n\alpha}$. Rescaling by $n$ ensures that minimizing the above loss is equivalent to maximizing the per-example MAP objective: $\frac{1}{n}[\log p(y|x,w,V) + \log p(w,V)]$.

We use Shwartz-Ziv et al. (2022)'s SimCLR Resnet-50 pre-trained initialization to initialize weights $w$.

***LearnedPriorIso.*** For *LearnedPriorIso*, we minimize the following MAP objective

$$L(w,V) := -\frac{1}{n}\sum_{i=1}^{n}\log p(y_i|x_i,w,V) + \frac{\alpha}{2}||w-\mu||^2 + \frac{\alpha}{2}||\text{vec}(V)||^2 \tag{7}$$

so that the weight decay search space is the same as SGD in PyTorch. We use Shwartz-Ziv et al. (2022)'s SimCLR Resnet-50 pre-trained initialization as the mean $\mu$ and to initialize weights $w$.

***LearnedPriorLR.*** For *LearnedPriorLR*, we minimize the following MAP objective

$$L(w, V) := -\frac{1}{n} \sum_{i=1}^{n} \log p(y_i|x_i, w, V) - \frac{1}{n} \log \mathcal{N}(w|\mu, \lambda\Sigma) + \frac{\alpha}{2}||\text{vec}(V)||^2. \tag{8}$$

Although not discussed in Shwartz-Ziv et al. (2022), they add $\varepsilon$ to the prior variance (they set the default value of $\varepsilon$ to 0.1). We include the addition of $\varepsilon$ to the prior variance by adding $\frac{\lambda}{2}\Sigma_{\text{diag}} + \varepsilon I$. We note that adding $\varepsilon$ to the prior variance has a significant effect if the covariance scaling factor is small (e.g., $\lambda = 1$).

## B  Side-by-side Experimental Settings

Table B.1: A detailed comparison of Shwartz-Ziv et al. (2022)'s and our experimental procedures.

| | **Shwartz-Ziv et al. (2022)'s** | **Ours** |
|---|---|---|
| Normalization | In their code, they use the mean and standard deviation of the entire training dataset. | We use the mean and standard deviation of the training dataset of size $n$. |
| Data augmentation | In their paper, they say they use random cropping and random horizontal flips. In their code, they use random vertical flips for some datasets as well. | We follow their paper, we use random cropping and random horizontal flips. |
| Batch size | They use a batch size of 128. | We use a batch size of 128. |
| Optimizer | They use SGD with a Nesterov momentum parameter of 0.9 for optimization. | We use SGD with a Nesterov momentum parameter of 0.9 for optimization. |
| Gradient clipping | In their paper, they do not mention gradient clipping. In their code, gradients are clipped so that the L2 norm is 2.0 by default. | We do not use gradient clipping. In our experiments, gradient clipping had little impact on performance. |
| Number of steps | For Bayesian inference they train for 30,000 steps using a cyclic learning rate (Zhang et al., 2019) with 5 chains. They never mention the number of steps they use for SGD-based methods. | We train for 6,000 steps using a cosine annealing learning rate (Loshchilov & Hutter, 2016) to reproduce their Bayesian inference settings for our SGD-based methods. |
| Initial learning rate | They select the initial learning rate from 7 logarithmically spaced learning rates between 0.1 and 0.0001. | We select the initial learning rate from 4 logarithmically spaced learning rates between 0.1 and 0.0001. We use a coarser search space to reduced computational complexity. We expect a finer search space to improve on our results. |
| Weight decay | They select weight decay from 7 logarithmically spaced values between $1e^{-6}$ and $1e^{-2}$, as well as without weight decay. They also divide weight decay by learning rate. | We select weight decay from 5 logarithmically spaced values between $1e^{-6}$ and $1e^{-2}$, as well as without weight decay. We do not divide weight decay by learning rate. We use a coarser search space to reduced computational complexity. We expect a finer search space to improve on our results. |
| | | Continued on next page. |

Table B.1: Continued from previous page.

| | **Shwartz-Ziv et al. (2022)'s** | **Ours** |
|---|---|---|
| Covariance scaling factor | In their paper, they plot test error on CIFAR-10 for 10 logarithmically spaced covariance scaling factors between $1e^0$ and $1e^9$ (see their Fig. 2d) but don't specify a search space for the hyperparameter. We assume they don't tune the covariance scaling factor since no search space is mentioned. | We select the covariance scaling factor from 10 logarithmically spaced values between $1e^0$ to $1e^9$. |
| Validation set size | In their paper, they don't say. | We use 1/5 of the training set. |
| Hyperparameter selection metric | In their paper, they don't say. In their code, they use loss. | We use the validation set NLL. |
| Rank of low-rank covariance | In their paper, they plot test error on CIFAR-10 and CIFAR-100 for covariance ranks between 0 and 9 (see their Fig. 2b) but use a rank of 5 for experiments. | We use a rank of 5 for experiments. |
| Prior epsilon (see App. A.2) | In their paper, they do not mention prior epsilon. In their code, the default is 0.1. | We use 0.1. |
| Covariance scaling | In their paper, they say to scale the covariance by multiplying $\lambda\Sigma$. In their code, they scale the covariance diagonal by multiplying $\lambda\Sigma_{\text{diag}}$ and the covariance factor by multiplying $\lambda Q$ which creates a multivariate normal distribution with covariance matrix $\lambda\Sigma_{\text{diag}} + \lambda^2\Sigma_{\text{LR}}$. They absorb the $\frac{1}{2}$ into $\lambda$. | We scale the covariance factor by multiplying $\sqrt{\lambda}Q$ which creates a multivariate normal distribution with covariance matrix $\frac{\lambda}{2}\Sigma_{\text{diag}} + \frac{\lambda}{2}\Sigma_{\text{LR}}$. |
| Double prior over weights $w$ | In their paper, they say to add a zero-mean isotropic Gaussian prior over added parameters (e.g., classification head) to solve the target task. In their code, they use weight decay over all parameters. | We do not use weight decay over all parameters. |

## C  NLL Results

Table C.1: CIFAR-10 heldout NLL (lower is better) as target train set size $n$ increases. For each method and train set size $n$, we ran our two-phase training pipeline (including hyperparameter tuning) on 3 distinct training sets (independent samples of $n$ examples from the provided full training set). We report mean NLL on test set across these 3 trials, along with the min-max range. 10 classes, balanced. See App. A for details.

| Method | $n = $ **10 (1/cl.)** | **100 (10/cl.)** | **1000 (100/cl.)** | **10000 (1k/cl.)** | **50000 (5k/cl.)** |
|---|---|---|---|---|---|
| StdPrior fromImgNet | 2.47 (2.10-3.20) | 1.48 (1.27-1.75) | 0.71 (0.66-0.79) | 0.23 (0.23-0.23) | 0.11 (0.10-0.11) |
| LearnedPriorIso fromImgNet | 2.31 (2.24-2.35) | 1.40 (1.30-1.50) | 0.68 (0.62-0.78) | 0.18 (0.18-0.19) | 0.09 (0.09-0.09) |
| LearnedPriorLR fromImgNet | 2.69 (2.30-3.39) | 1.31 (1.20-1.44) | 0.58 (0.56-0.58) | 0.22 (0.21-0.23) | 0.10 (0.10-0.11) |

Table C.2: Oxford Flowers heldout NLL (lower is better) as target train set size $n$ increases. For each method and train set size $n$, we ran our two-phase training pipeline (including hyperparameter tuning) on 3 distinct training sets (independent samples of $n$ examples from the provided full training set). We report mean NLL on test set across these 3 trials, along with the min-max range. 102 classes, balanced. See App. A for details.

| Method | $n = 102$ (1/cl.) | 510 (5/cl.) | 1020 (10/cl.) |
|---|---|---|---|
| StdPrior fromImgNet | 3.69 (3.40-4.23) | 1.00 (0.96-1.05) | 0.59 (0.58-0.61) |
| LearnedPriorIso fromImgNet | 4.43 (3.62-5.14) | 1.00 (0.96-1.03) | 0.58 (0.57-0.59) |
| LearnedPriorLR fromImgNet | 4.00 (3.59-4.81) | 1.04 (0.95-1.14) | 0.54 (0.52-0.58) |

Table C.3: Oxford-IIIT Pets heldout NLL (lower is better) as target train set size $n$ increases. For each method and train set size $n$, we ran our two-phase training pipeline (including hyperparameter tuning) on 3 distinct training sets (independent samples of $n$ examples from the provided full training set). We report mean NLL on test set across these 3 trials, along with the min-max range. 37 classes, balanced. See App. A for details.

| Method | $n = 37$ (1/cl.) | 370 (10/cl.) | 3441 (93/cl.) |
|---|---|---|---|
| StdPrior fromImgNet | 3.33 (3.25-3.40) | 1.78 (1.72-1.83) | 0.59 (0.55-0.66) |
| LearnedPriorIso fromImgNet | 3.62 (3.56-3.69) | 1.79 (1.71-1.87) | 0.58 (0.52-0.66) |
| LearnedPriorLR fromImgNet | 3.62 (3.59-3.63) | 1.69 (1.64-1.72) | 0.52 (0.52-0.53) |

Table C.4: FGVC-Aircraft heldout NLL (lower is better) as target train set size $n$ increases. For each method and train set size $n$, we ran our two-phase training pipeline (including hyperparameter tuning) on 3 distinct training sets (independent samples of $n$ examples from the provided full training set). We report mean NLL on test set across these 3 trials, along with the min-max range. 100 classes, balanced. See App. A for details.

| Method | $n = 100$ (1/cl.) | 500 (5/cl.) | 1000 (10/cl.) | 5000 (50/cl.) |
|---|---|---|---|---|
| StdPrior fromImgNet | 4.88 (4.71-5.18) | 3.52 (3.43-3.57) | 2.46 (2.40-2.50) | 0.61 (0.60-0.62) |
| LearnedPriorIso fromImgNet | 4.67 (4.66-4.68) | 3.37 (3.23-3.63) | 2.01 (1.94-2.05) | 0.57 (0.56-0.57) |
| LearnedPriorLR fromImgNet | 4.68 (4.66-4.71) | 3.37 (3.33-3.45) | 2.00 (1.96-2.03) | 0.64 (0.60-0.67) |

Table C.5: HAM10000 NLL on heldout data (lower is better) as target train set size $n$ increases. For each method and size $n$, we ran our two-phase training (including hyperparameter tuning) on 3 distinct training sets (independent samples of size $n$ from the provided full training set). We report mean NLL on test set across 3 trials (min-max range). 4 classes, ranging from 11%-67% prevalance. See App. A for details.

| Method | $n = 100$ | 1000 |
|---|---|---|
| StdPrior fromImgNet | 1.16 (0.91-1.33) | 0.90 (0.79-1.00) |
| LearnedPriorIso fromImgNet | 1.16 (0.91-1.32) | 0.83 (0.79-0.91) |
| LearnedPriorLR fromImgNet | 1.35 (0.89-1.90) | 0.89 (0.78-1.09) |

# D Comparison of Fully-Supervised Pretraining

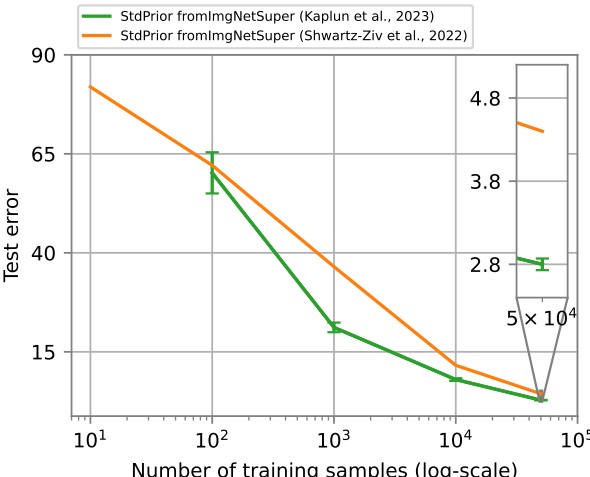

Figure D.1: Error rate (lower is better) vs. target train set size on CIFAR-10, for standard transfer learning from ImageNet using fully-supervised pre-training. The orange line are results copied from Shwartz-Ziv et al. (2022) (their Tab. 2). The green line is a third-party experiment copied from Kaplun et al. (2023). Plotted mean and standard deviations confirmed via direct correspondence with Kaplun et al.. **Takeaway: In third-party experiments, standard transfer learning (*StdPrior*) performs better at dataset sizes $n \in \{1000, 10000, 50000\}$ than reported in Shwartz-Ziv et al. (2022).**

# E    Parameter Interpolation

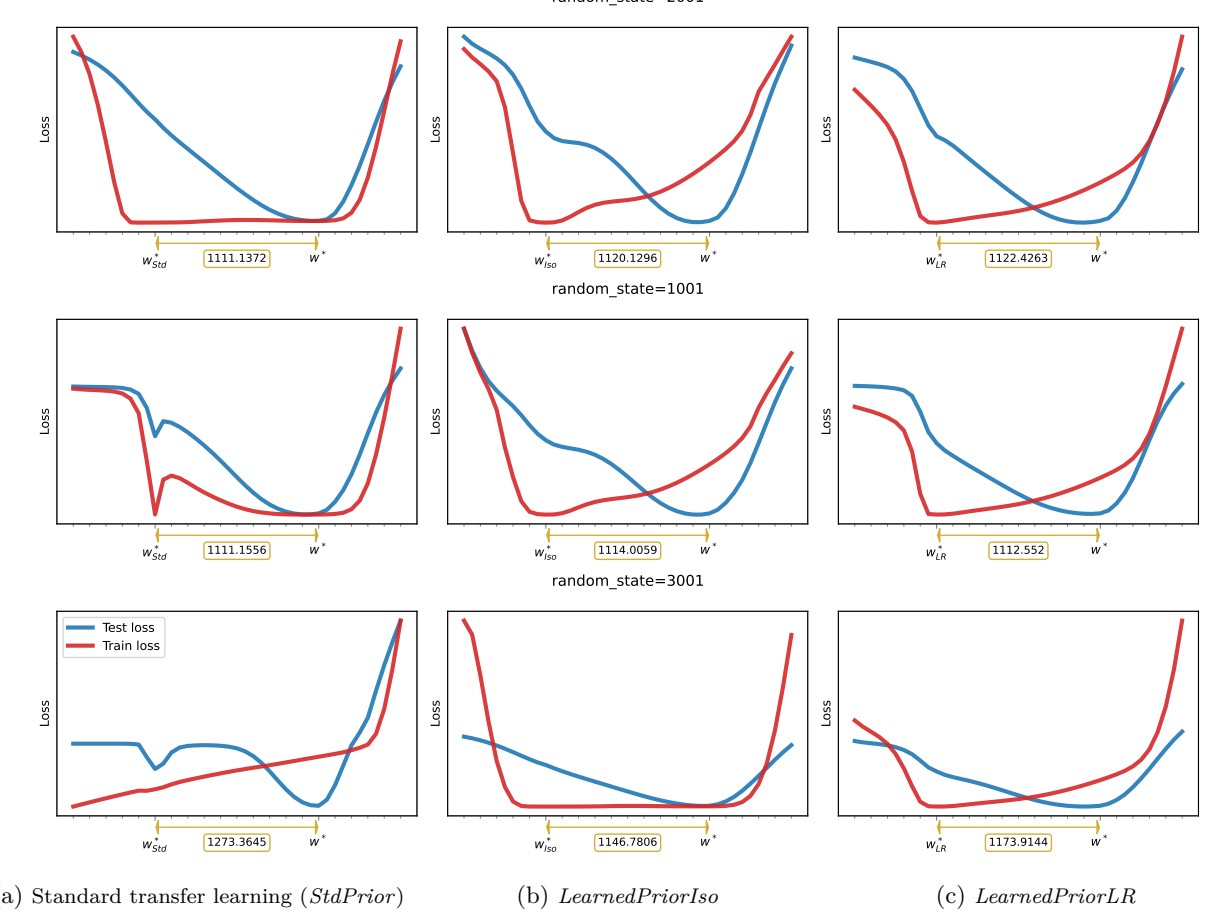

(a) Standard transfer learning (*StdPrior*)          (b) *LearnedPriorIso*          (c) *LearnedPriorLR*

Figure E.1: Expanded version of Fig. 2, including a third column for the *LearnedPriorIso* method. *Each panel:* We assess a 1D slice of the high-dimensional landscape by linearly interpolating parameters $w$ between the optima $w^*_{Std}$ found via minimizing the standard MAP objective (left), the optima $w^*_{Iso}$ found via minimizing the *LearnedPriorIso* MAP objective (center), or the optima $w^*_{LR}$ found via minimizing the *LearnedPriorLR* MAP objective (right) and the optima $w^*$ found at the largest dataset size. Red curve shows the indicated training loss (varies by column) on CIFAR-10 with $n = 1000$ samples; blue curve (varies by column) shows CIFAR-10 test set NLL. Each row shows results from a different train set sample of size $n = 1000$. The gap between the optima found via training and the test set's ideal minimum is shown as a double-sided gold arrow.

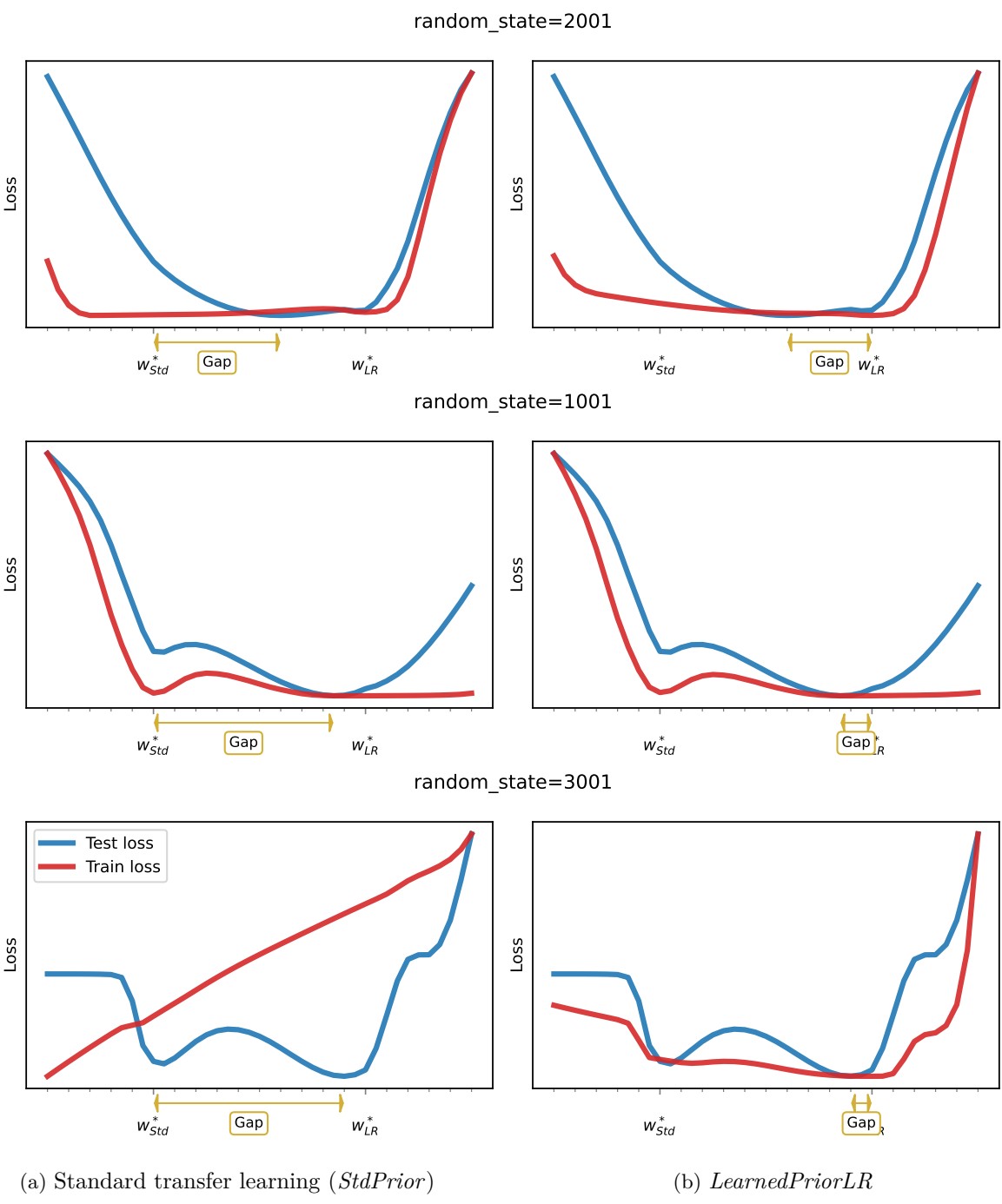

(a) Standard transfer learning ($StdPrior$)   (b) $LearnedPriorLR$

Figure E.2: Alternative version of Fig. 2, looking at a 1D slice that interpolates between the optima found by each method instead of the optima found by each method and the optima found at the largest dataset size. *Each panel:* We assess a 1D slice of the high-dimensional landscape by linearly interpolating parameters $w$ between the optima $w^*_{Std}$ found via minimizing the standard MAP objective and optima $w^*_{LR}$ found via minimizing the *LearnedPriorLR* MAP objective. Red curve shows the indicated training loss (varies by column) on CIFAR-10 with $n = 1000$ samples; blue curve (same across columns) shows CIFAR-10 test set NLL. Each row shows results from a different train set sample of size $n = 1000$. The gap between the optima found via training and the test set's minimum on the 1D slice is shown as a double-sided gold arrow.

## F   Comparison with Full Bayesian Inference

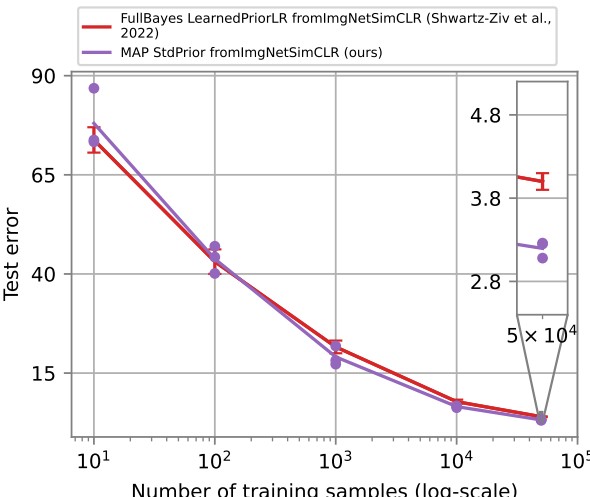

Figure F.1: Error rate (lower is better) vs. target train set size on CIFAR-10, for standard transfer learning and full Bayesian inference with *LearnedPriorLR* from ImageNet. The red line are results copied from Shwartz-Ziv et al. (2022) (their Tab. 10). **Takeaway: Our standard transfer learning (*StdPrior*) results on CIFAR-10 are as good as or better than the full Bayesian inference results reported in Shwartz-Ziv et al. (2022).**

## G   Hyperparameter Sensitivity

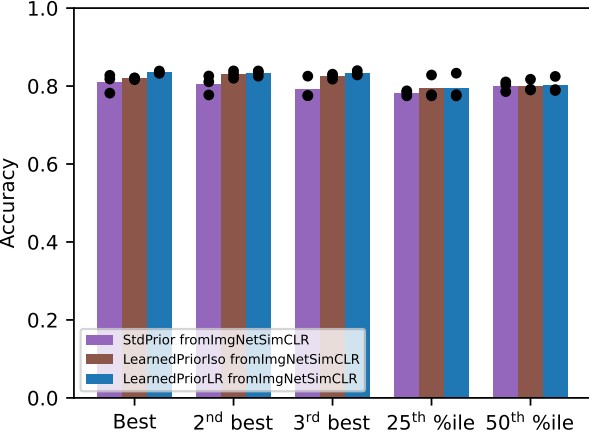

Figure G.1: Sensitivity of selected hyperparameters for CIFAR-10 $n = 1000$. We report CIFAR-10 heldout accuracy (higher is better) with various selected hyperparameters as validation NLL increases.

