# OpenReview forum: "Transfer Learning with Informative Priors: Simple Baselines Better than Previously Reported"
_TMLR — Accepted by TMLR_

### Review · Reviewer_KTv2 · 2024-03-09

**Summary Of Contributions:**

The paper studies the empirical performance of using informed prior in transfer learning. The primary method of focus is [1] who proposed that instead of learning a point estimate of a good initialization from pretraining data, one could learn a distribution of weights in the form of mean and a low-rank covariance and perform the finetuning is a Bayesian manner. [1] found that their proposed Bayesian finetuning method performed significantly better than the baseline which is simply initializing the backbone at $\mu$ which is the weights of the pretrained model and apply standard $\ell_2$ regularization during finetuning / transfering. However, this work compare the performance of [1] against a naive baselinea and a baseline that only use isotropic posterior without the low-rank covariance with all their hyperparamters properly tuened. The result shows that the naive baseline can perform much better if the hyperparameters are properly tuned. In fact, there is almost no gap between the naive baseline and that of [1] which raises the question about whether the added complexity is actually needed. Furthermore, the paper provides a counter example to the loss-alignment hypothesis.

**Reference**

[1] Pre-Train Your Loss: Easy Bayesian Transfer Learning with Informative Priors. Shwartz-Ziv et al.

**Audience:**

Yes

**Broader Impact Concerns:**

I do not see any concerns on the ethical implications.

**Claims And Evidence:**

Yes

**Requested Changes:**

If possible, I would like to see a visualization / analysis of how the performance varies as a function of the hyperparameters (e.g., some kind of sensitivity analysis) for all methods. I feel like something like that would be beneficial to further understanding the strength and weaknesses of the studied methods.

**Strengths And Weaknesses:**

**strengths**

- The paper uses careful experimental design to independently verify the claims made by another paper and reaches more nuanced conclusion. This sort of work is very important and perhaps not as valued as they should be.
- The paper is carefully written and very easy to follow and presents a balanced view of the issue of prior works.

**weaknesses**

- I do not see any major flaws in the paper.

---

> ### Author Response · Authors · 2024-04-09
> **Response to Reviewer KTv2**
>
> Thank you for your thoughtful and encouraging feedback! We agree that this sort of work is important and perhaps undervalued. We have included a sensitivity analysis of the selected hyperparameters for each method on CIFAR-10 at $n=1000$ in App. G. The main takeaway message is that performance is similar for the best 3 hyperparameters across all methods.

---

### Review · Reviewer_uKdC · 2024-03-24

**Summary Of Contributions:**

The paper investigates the performance of the LearnedPriorLR method of Shwartz-Ziv et al. (2022) in the context of transfer learning. The paper finds that the method does not have a performance difference with the baseline, as claimed in the original paper, and that the alignment of the loss surfaces of the train and test data is not as good as claimed.

**Audience:**

Yes

**Claims And Evidence:**

No

**Requested Changes:**

* Adopting the broader range of benchmarks used in the (dense) fine-tuning literature (e.g., from Li et al. (2018) or Gouk et al. (2021)) would allow for a more reliable set of results.
* The experiment investigating the alignment loss surfaces should be rethought. A more reliable way to investigate the alignment of the loss surfaces would be to consider the distance between the train and test optima in the full parameter space, rather than on a one dimensional slice.
* It would be nice if the experiments could be re-run with a more uniform hyperparameter optimisation procedure across all methods considered, and using the validation accuracy rather than NLL.

Li et al. Explicit inductive bias for transfer learning with convolutional networks. In ICML,2018.

Gouk et al. Distance-Based Regularisation of Deep Networks for Fine-Tuning. In ICLR, 2021.

**Strengths And Weaknesses:**

Strengths:
* Machine Learning is going through a reproducibility crisis, due to the lack of reliable experimental setups and abundance of overclaiming, so paper is definitely of interest to someone in the TMLR community. Given the topic of transfer learning is currently very popular, and the method under consideration is growing in popularity, I think the paper is very well motivated.
* The paper is well written. It provides a good introduction to the problem formulation of the class of methods considered and clearly explains the scope of the claims being made.
* The paper does a good job of designing an experimental setup that mimics the setting considered by Shwartz-Ziv et al. (2022), with extensive effort to exam why the results are different given in the appendix.
* Findings 1 and 2 are interesting and have practical significance to the research community.

Weaknesses:
* A limitation of the work in Shwartz-Ziv et al. (2022) is the lack of fine-tuning tasks considered in the evaluation. This weakness is carried to the present work, and means that the conclusions drawn in this paper can only be relied upon with low confidence.
* The experiment investigating the alignment loss surfaces seems flawed. The one dimensional slice obtained by interpolating between two different optima is very unlikely to contain an optimum for the test data. Looking at the precise distance between the optima that is found and the optimum for the test data on this one dimensional slice will not be a good approximation of the actual distance between the train and test optima.

Some questions:
* For the repeated trials discussed in Section 3, are the replicate training sets sampled with or without replacement?
* Regarding hyperparameter tuning, why is a lower dimensioned space of hyperparamters search for LearnedPriorLR?
* Why is NLL used as the validation criterion? Typically this would be accuracy.

---

> ### Author Response · Authors · 2024-04-09
> **Response to Reviewer uKdC**
>
> > A limitation of the work in Shwartz-Ziv et al. (2022) is the lack of fine-tuning tasks considered in the evaluation. This weakness is carried to the present work, and means that the conclusions drawn in this paper can only be relied upon with low confidence.
>
> We have added results on another dataset (Oxford Flowers) at 3 different dataset sizes to address this weakness.  Our updated paper reports results on 4 datasets (totalling 13 different sizes). We would also like to stress that we are considering more settings than Li et al., (2018) who report results on 3 datasets (totalling 4 different sizes) and Gouk et al., (2021) who report results on 6 datasets (totalling 6 different sizes).
>
> > The experiment investigating the alignment loss surfaces seems flawed. The one dimensional slice obtained by interpolating between two different optima is very unlikely to contain an optimum for the test data. Looking at the precise distance between the optima that is found and the optimum for the test data on this one dimensional slice will not be a good approximation of the actual distance between the train and test optima.
> The experiment investigating the alignment loss surfaces should be rethought. A more reliable way to investigate the alignment of the loss surfaces would be to consider the distance between the train and test optima in the full parameter space, rather than on a one dimensional slice.
>
> We have taken your advice and report the Euclidean distance between the optima found by each method (StdPrior and LearnedPriorLR) at $n=1000$ and the optima found at $n=50000$.
>
> Since this Euclidean distance is also a 1D slice, we have included plots of the loss landscapes along it. We note that the optima found at $n=50000$ is not the “true” optima for the test set but is the best estimate we have as fitting a network to the test set to find the minimum would result in overfitting to the test set.
>
> We hope that this updated plot addresses your **concerns about Claims and Evidence**. If not, please let us know how we can further address this.
>
> > For the repeated trials discussed in Section 3, are the replicate training sets sampled with or without replacement?
>
> Replicate training sets are sampled without replacement. We have added this clarification to our paper.
>
> > Regarding hyperparameter tuning, why is a lower dimensioned space of hyperparamters search for LearnedPriorLR?
>
> LearnedPriorLR does not have a lower dimension hyperparameter search space. In the Hyperparameter tuning section of our paper we say that “LearnedPriorLR is allowed to search a larger hyperparameter space.” For LearnedPriorLR we search over separate hyperparameters for the covariance scaling factor $\lambda$ and the classification head weight decay $\tau$. For both StdPrior and LearnedPriorIso we fix $\lambda = \tau$ since both the backbone and classifier prior covariance are isotropic.
>
> We want to stress that the focus of our work is replication and that this is the hyperparameter search space outlined in Shwartz-Ziv et al. (2022).
>
> > Why is NLL used as the validation criterion? Typically this would be accuracy.
> It would be nice if the experiments could be re-run with a more uniform hyperparameter optimisation procedure across all methods considered, and using the validation accuracy rather than NLL.
>
> We used NLL as the validation criterion for three reasons.
> Firstly, NLL was used for the sake of reproducibility as this was the criterion used in  Shwartz-Ziv et al. (2022)  (see Tab. B.1 in App. B for a comparison of Shwartz-Ziv et al. (2022)’s and our experimental procedures).
> Secondly, as the training/validation set sizes grow smaller, accuracy becomes very sensitive to every misclassified image and becomes a widely fluctuating quantity. As a result, choosing the best set of hyperparameters becomes a difficult and unreliable task. NLL is less sensitive (e.g. it does not jump from 0 to 1 when the correct class no longer has predicted majority probability).
> Thirdly, selecting hyperparameters based on validation accuracy would not make sense when the number of training samples is less than or equal to the number of classes (there are numerous of these cases considered in Shwartz-Ziv et al. (2022) and thus our work too).
>
> We’re not sure exactly what could be "more uniform" about our tuning procedure, perhaps the reviewer could clarify. Perhaps what was meant is that our procedure allows LearnedPriorLR more hyperparameters to tune than StdPrior, again following past work. We think even with that, our results show that this advantage doesn’t substantially improve LearnedPriorLR over StdPrior in many cases, adding to our overall story that simpler baselines work better than previously reported.
>
> For these reasons, plus the substantial resource consumption required to rerun all experiments (each of the 4 datasets would require ~7 days of full-time GPU effort), we are satisfied with the current protocol and hope the reviewers are satisfied as well.

---

> ### Author Response · Authors · 2024-04-09
> **Response to Reviewer uKdC**
>
> > Adopting the broader range of benchmarks used in the (dense) fine-tuning literature (e.g., from Li et al. (2018) or Gouk et al. (2021)) would allow for a more reliable set of results.
>
> If by a "broader range of benchmarks," you are asking for experiments on additional datasets, we hope our new results on the Oxford Flowers dataset (see Tab 3 of our latest revision) helps address your concern.
>
> If instead you mean additional comparisons to other transfer learning methods, we would like to note that $L^2-SP$ which Li et al. (2018) recommend as "the standard baseline for solving transfer learning tasks and benchmarking new algorithms" (Li et al., 2018), is the same as what we call LearnedPriorIso. Both methods use the prior proposed in Chelba & Acero (2006) (see a detailed explanation in App. A.2). We have added this clarification to our paper.

---

### Review · Reviewer_M6Gg · 2024-03-27

**Summary Of Contributions:**

This paper conducts a replication study of the findings of Schwartz-Ziv et al. which suggested that learned Bayesian priors can improve sample-efficiency in transfer learning over point estimate initializations. It finds that the purported benefits of learned priors over simpler pre-training baselines are largely a result of poor tuning of the baselines, and that with appropriate tuning these simple baselines exhibit minimal reductions in performance relative to, and in some cases even exceed the performance of, the method of Schwartz-Ziv et al. This analysis is extended to contrast low-rank and isotropic prior covariance matrices on a variety of tasks, including HAM10000, which was not considered by Schwartz-Ziv et al. Finally, the paper also studies the loss landscapes of networks trained with learned priors, visualizing the alignment between train and test losses in a one-dimensional region of parameter space obtained by interpolating between two local minima.

**Audience:**

Yes

**Broader Impact Concerns:**

No broader impact concerns.

**Claims And Evidence:**

Yes

**Requested Changes:**

See "weaknesses" above. I also would like to see the following more minor questions addressed.

- Does the pre-training procedure exactly match that of Schwartz-Ziv et al.?
- How does the experimental protocol used to generate Figure 1 differ from Table 2? Or are they based on the same data with Table 2 incorporating a second baseline (LearnedPriorIso)?
- Why are error bars not provided in Figure 1? Am I correct in interpreting this as only presenting two seeds per dataset size? If so, how does this correspond to the claim that three different randomly selected datasets were used in the evaluations?
- It would be useful to indicate what the different rows correspond to in the plot titles in addition to the description in the legend of Figure 2.

**Strengths And Weaknesses:**

**Strengths:**

- The paper makes a concerted effort to produce a clear and reproducible replication of the procedure of Schwartz-Ziv et al. I particularly appreciated the table in the appendix which included a description of key factors where the empirical choices in this work differed from those of the original paper proposing learned priors.
- The claim that “ standard transfer learning without any informative prior performs better than reported in Shwartz-Ziv et al. (2022).” is well-supported by Figure 1, at least with respect to the CIFAR-10 dataset.
- The paper performs some nice analysis which fleshes out the claimed mechanism by which learned priors help improve performance via alignment between train and test loss landscapes actually occurs when using learned priors. In my experience with the Bayesian Deep Learning literature, it is very common to claim a particular mechanism (usually motivated by some connection to Bayesian inference) is driving the success of a method without rigorously testing whether this is the case, and I appreciate the paper’s work aiming to falsify these claims empirically.
- The paper is very well-written and clearly describes and justifies its methodology.


**Weaknesses**

- Schwartz-Ziv et al. evaluate their method on 4 benchmarks in their Figure 3. Of these, only two are evaluated in this paper. In order to be fully convinced of the claim that learned priors do not make a difference to performance, I would want to see the analysis of Figure 1 replicated for each of these datasets.
- The claim that learned priors do not improve alignment of the loss landscape requires additional evidence. Figure 2 seems to show very good alignment, in the sense that the train and test losses roughly agree on local minima. Figure 1 of Schwartz-Ziv et al. shows examples where a local minimum of the train loss is far from a minimum of the test loss.
  - I am also not sure if the choice of endpoints of the linear interpolation necessarily makes the most sense. If I understand correctly, the interpolation is performed between the minima found by the standard and learned prior MAP objectives. I think in the idealized figure 1 from Schwartz-Ziv et al. the visualization was intended to be from the initial parameters of the fine-tuning phase.
    - Given that the authors already have the experimental aparatus for evaluating loss landscapes, I would suggest contrasting the loss landscapes of the learned prior with those obtained by regular L2 regression as well as a learned prior with isotropic covariance (and for good measure perhaps also using the low-rank covariance matrix with a zero-centered prior), so that we can not just see whether the loss landscapes are aligned but whether they are more aligned in one method compared to another.
- One difference between this paper and that of Schwartz-Ziv is the use of SGD in lieu of HMC when training with learned priors. I would assume that a learned prior would shine more when used with a Bayesian training method, and so could believe that some of the effect that is observed in this paper might be attributed to the use of SGD as opposed to HMC. Can the authors comment on this choice?

---

> ### Author Response · Authors · 2024-04-09
> **Response to Reviewer M6Gg**
>
> > Schwartz-Ziv et al. evaluate their method on 4 benchmarks in their Figure 3. Of these, only two are evaluated in this paper. In order to be fully convinced of the claim that learned priors do not make a difference to performance, I would want to see the analysis of Figure 1 replicated for each of these datasets.
>
> We have made two changes to address this weakness. First, we have added results on another dataset (Oxford Flowers) at 3 different dataset sizes (we are actively running further experiments on another dataset, hopefully ready in a few days time). Second, we have softened Finding 2 from "informative priors are not convincingly better than standard transfer learning" to "informative priors perform slightly better than standard initialization-only transfer learning but never by a substantial margin (by 3 or more percentage points)" because of the more noticeable performance improvement after fixing two bugs we inherited from Shwartz-Ziv et al. (2022)’s implementation.
>
> > The claim that learned priors do not improve alignment of the loss landscape requires additional evidence. Figure 2 seems to show very good alignment, in the sense that the train and test losses roughly agree on local minima. Figure 1 of Schwartz-Ziv et al. shows examples where a local minimum of the train loss is far from a minimum of the test loss.
> I am also not sure if the choice of endpoints of the linear interpolation necessarily makes the most sense. If I understand correctly, the interpolation is performed between the minima found by the standard and learned prior MAP objectives. I think in the idealized figure 1 from Schwartz-Ziv et al. the visualization was intended to be from the initial parameters of the fine-tuning phase.
> Given that the authors already have the experimental aparatus for evaluating loss landscapes, I would suggest contrasting the loss landscapes of the learned prior with those obtained by regular L2 regression as well as a learned prior with isotropic covariance (and for good measure perhaps also using the low-rank covariance matrix with a zero-centered prior), so that we can not just see whether the loss landscapes are aligned but whether they are more aligned in one method compared to another.
>
> We have updated our 1D plot by interpolating between the optima found by each method (StdPrior and LearnedPriorLR) at $n=1000$ and the optima found at $n=50000$. We have also included the Euclidean distance between the optimas. We note that the optima found at $n=50000$ is not the "true" optima for the test set but is the best estimate we have as fitting a network to the test set to find the minimum would result in overfitting to the test set.
>
> We agree with you that plotting the loss landscape of LearnedPriorIso on this 1D slice as well can provide further insight into the alignment of each method. We have included a 3rd column in this plot in App. E.
>
> > One difference between this paper and that of Schwartz-Ziv is the use of SGD in lieu of HMC when training with learned priors. I would assume that a learned prior would shine more when used with a Bayesian training method, and so could believe that some of the effect that is observed in this paper might be attributed to the use of SGD as opposed to HMC. Can the authors comment on this choice?
>
> Shwartz-Ziv et al. (2022) propose two new methods BNN Learned Prior and SGD Learned Prior. They use SGHMC for BNN Learned Prior but use SGD for SGD Learned Prior. In our paper, we focus on replicating Shwartz-Ziv et al. (2022)’s work using SGD Learned Prior, primarily because (1) we view this as the simplest setting most likely to be adopted by practitioners, and (2) the results reported in Shwartz-Ziv et al. (2022) suggest that large gains are possible even with SGD Learned Prior (as we write in our introduction).
>
> > Does the pre-training procedure exactly match that of Schwartz-Ziv et al.?
>
> Yes. We do not do any pre-training ourselves but directly use the priors released by Schwartz-Ziv et al. (2022). This is reported in the "Common source task knowledge" paragraph of Sec. 3.
>
> > How does the experimental protocol used to generate Figure 1 differ from Table 2? Or are they based on the same data with Table 2 incorporating a second baseline (LearnedPriorIso)?
>
> The results in Figure 1 are the same as in Table 2 (reported in terms of error instead of accuracy). In Figure 1, we focus on comparing performance directly with Shwartz-Ziv et al. (2022). In Table 2, we add a second baseline (LearnedPriorIso).

---

> ### Author Response · Authors · 2024-04-09
> **Response to Reviewer M6Gg**
>
> > Why are error bars not provided in Figure 1? Am I correct in interpreting this as only presenting two seeds per dataset size? If so, how does this correspond to the claim that three different randomly selected datasets were used in the evaluations?
>
> Instead of adding error bars to Figure 1, we plot dots for all three random seeds. We think this helps readers get more direct insight into how performance varies across replicates. We have updated Figure 1 so that it is clearer that there are three random seeds.
>
> > It would be useful to indicate what the different rows correspond to in the plot titles in addition to the description in the legend of Figure 2.
>
> Thanks for the suggestion. We have added a subtitle to each row indicating the random state used for generating the replicate training set.

---

### Author Response · Authors · 2024-04-09
**General Response to Reviewers and Action Editor**

We would like to thank all the reviewers for their thoughtful feedback.

Here we summarize the contributions and strengths highlighted by reviewers as well as discuss broad changes to the paper.

The reviewers mentioned that this sort of replication work is very important (KTv2) and of interest to the TMLR community (uKdC). All reviewers agreed the paper is well written (KTv2, uKdC, M6Gg) and does a good job of designing an experimental setup that mimics the setting considered by Shwartz-Ziv et al. (2022) (KTv2, uKdC, M6Gg). Reviewers mentioned that Findings 1 and 2 are interesting and have practical significance to the research community (uKdC) and Finding 3 is a nice analysis of the previously claimed mechanism (M6Gg).

We have made three major changes to address weaknesses raised by reviewers (updated text is red).

## Major Change 1: Add experiments on Oxford Flowers dataset

First, we have added results on the Oxford Flowers dataset (Nilsback & Zisserman, 2008) a dataset included in previous experiments by Shwartz-Ziv et al. (2022). For Oxford Flowers experiments, we modify prior experiments following the same protocols we used in our other experiments:
*  ensure each class is equally represented in the training data, that at least one sample from each class is included in the training data
*  we use a validation set is proportional with the training set

Shwartz-Ziv et al. (2022) validate on the predefined validation set specified by the dataset creators. In our minds, it is not realistic to train on a range of dataset sizes of 102 or 510 images but validate on 1020 images. Practitioners looking for guidance in how to use transfer learning on their dataset will be unlikely to put more images in validation than training. Instead, we ignore the predefined validation set and use 1/5 of the training set for our two-phase training pipeline.

In our Oxford Flowers results, we see that StdPrior is 3 percentage points better at dataset size $n=510$ compared to results reported at $n=500$ in Shwartz-Ziv et al. (2022), essentially erasing the gap in performance previously attributed to informative priors. At the largest dataset size ($n=1020$), we see a decrease in performance for all our methods compared to Shwartz-Ziv et al. (2022). This is likely because we ignore the validation set specified by the creators and instead use 1/5 of the training set for our two-phase training pipeline. This is a more realistic experiment setting as if we had access to a large validation set we would likely validate on less images and use more images for training.

## Major Change 2: Bug fixes for competitor methods lead to updates to Finding 2

Additionally, in our own careful review of the code we inherited from Shwatz-Ziv et al. (2022) to run their method, we recently identified an issue with correctness in their implementation of rescaling the low-rank covariance matrix. We also removed the redundant "double prior" due to their code’s use of weight decay. Details for both of these can be found in Tab. B.1 in App. B.

We fixed both these issues and reran all experiments for their method (LearnedPriorLR). The updated LearnedPriorLR results have slightly improved performance in several settings, more noticeable than before. However, we still cannot find a setting where LearnedPriorLR improves by 3 or more percentage points.

Due to these bug fixes, we have softened Finding 2 from "informative priors are not convincingly better than standard transfer learning" to "informative priors perform slightly better than standard initialization-only transfer learning but never by a substantial margin (by 3 or more percentage points)."

## Major Change 3: Improved analysis of loss landscape in Figure 2

Finally, we have updated Figure 2 by interpolating between the optima found by each method (StdPrior and LearnedPriorLR) at $n=1000$ and the optima found at $n=50000$. We have also included the Euclidean distance between the optimas. This suggestion was made by Reviewer uKdC.

## Anticipated change: More experiments

We are also currently running experiments on the Fine-Grained Visual Classification of Aircraft (FGVC-Aircraft) dataset (Maji et al., 2013) which will hopefully be finished by Wednesday, April 10th. We think this is a more interesting test case for transfer learning than CIFAR-100, the other dataset studied by Shwartz-Ziv et al. (2022).

We hope reviewers can appreciate that running a full slate of experiments on each dataset requires ~7 full days of compute.

---

### Author Response · Authors · 2024-04-17
**General Response to Reviewers and Action Editor**

We added results on the Fine-Grained Visual Classification of Aircraft (FGVC-Aircraft) dataset (Maji et al., 2013). These results show substantial gains (>8 points) for methods using informative priors. We have updated our paper's story to talk about the variation in relative gains of methods using informative priors over standard transfer learning across datasets (updated text is purple). We hope that the addition of Oxford Flowers (at 3 different dataset sizes) and FGVC-Aircraft (at 4 different dataset sizes) addresses reviewers concerns about the lack of target tasks.

---

### Decision · Action_Editor_PvxG · 2024-05-06

**Recommendation:** Accept as is

**Comment:**

All reviewers were leaning accept and two out of three reviewers recommended the paper for a reproducibility certification.

**Audience:**

Readers who have shown an interest in the original work by Shwartz-Ziv et al. will certainly be interested in this study, which contributes additional insights to the discourse on priors in transfer learning.

**Claims And Evidence:**

The reviewers agree that this is a carefully executed reproducibility study of the work by Shwartz-Ziv et al., which offers novel insights into the actual benefits of this approach.